# The ubiquitin-like modifier FAT10 is degraded by the 20S proteasome in vitro but not in cellulo

Franziska Oliveri[1],*⬝, Steffen Johannes Keller[1],*⬝, Heike Goebel[1], Gerardo Omar Alvarez Salinas[1], Michael Basler[1,2]⬝

Ubiquitin-independent protein degradation via the 20S proteasome without the 19S regulatory particle has gained increasing attention over the last years. The degradation of the ubiquitin-like modifier FAT10 by the 20S proteasome was investigated in this study. We found that FAT10 was rapidly degraded by purified 20S proteasomes in vitro, which was attributed to the weak folding of FAT10 and the N-terminally disordered tail. To confirm our results in cellulo, we established an inducible RNA interference system in which the AAA-ATPase Rpt2 of the 19S regulatory particle is knocked down to impair the function of the 26S proteasome. Using this system, degradation of FAT10 in cellulo was strongly dependent on functional 26S proteasome. Our data indicate that in vitro degradation studies with purified proteins do not necessarily reflect biological degradation mechanisms occurring in cells and, therefore, cautious data interpretation is required when 20S proteasome function is studied in vitro.

## Introduction

The ubiquitin–proteasome system is the primary route of protein degradation in eukaryotic cells which uses ubiquitin tags as signals for proteins to be degraded by the 26S proteasome. This process is involved in virtually all cellular processes and therefore of critical importance to cells. The 26S proteasome consists of the catalytically active 20S core particle (CP) and one or two 19S regulatory particles (RPs) (reviewed in the study by Glickman and Ciechanover [2002]; Goldberg [2003]; Schwartz and Ciechanover [2009]; Schmidt and Finley [2014]). During the last years, it was reported that more than 20% of all proteins can be degraded independently of ubiquitin and the 19S RP and are directly processed by the free 20S CP (Baugh et al, 2009; Ben-Nissan & Sharon, 2014; Kumar Deshmukh et al, 2019). This direct degradation seems to be important for the clearance of oxidized proteins (Aiken et al, 2011; Pickering & Davies,

2012), neuronal communication (Ramachandran & Margolis, 2017), and post-translational processing (Baugh & Pilipenko, 2004; Sorokin et al, 2005; Moorthy et al, 2006; Gao et al, 2010; Morozov et al, 2017; Solomon et al, 2017). It could be shown that during oxidative stress, the 26S proteasome disassembles into the 20S core and the 19S cap (Aiken et al, 2011). Moreover, intrinsic disordered proteins (IDPs) are reported to be degraded directly by the 20S proteasome and do not require ubiquitin tagging (Asher et al, 2005, 2006; Tsvetkov et al, 2008; Baugh et al, 2009; Hwang et al, 2011; Ben-Nissan & Sharon, 2014; Solomon et al, 2017; Myers et al, 2018). Many of these studies used cell lysates and isolated proteins for in vitro studies and mutations of the ubiquitin acceptor sites and proteasome inhibitors to demonstrate ubiquitin-independent degradation (reviewed in the study by Hwang et al [2011]; Ben-Nissan and Sharon [2014]; Kumar Deshmukh et al [2019]). However, studying 20S proteasome–mediated protein degradation in cellulo is rather limited because both proteasome forms, the 26S and the 20S, are very abundant in all cells. Therefore, one aim of our study was the establishment of a system with impaired 19S RP function to study the 20S proteasome–mediated degradation in cellulo. Our approach uses RNAi-mediated knockdown of the AAA-ATPase Rpt2, one of the 19S RP subunits, that is critically important for gate opening and stabilization of the 26S complex (Köhler et al, 2001; Smith et al, 2007; Gillette et al, 2008; Rabl et al, 2008).

FAT10 (human leukocyte antigen-F adjacent transcript 10 or ubiquitin D) is a ubiquitin-like modifier, that is expressed in tissues of the immune system and upon inflammation in mammals (Liu et al, 1999; Raasi et al, 1999; Lukasiak et al, 2008; Choi et al, 2014; Mah et al, 2019). Mediated by the di-glycine motif at the C-terminus (Fan et al, 1996), FAT10 can be covalently attached to proteins by a conserved enzymatic cascade similar to ubiquitin (summarized in the study by Aichem and Groettrup [2020]). Therefore, it can directly target proteins for degradation, independent of ubiquitin (Schmidtke et al, 2009, 2014). FAT10 itself is rather short-lived and rapidly degraded in vitro and in cellulo in a ubiquitin-independent manner (Hipp et al, 2005; Schmidtke et al, 2009; Aichem et al, 2014). Its degradation does not require the segregase valosin-containing protein (VCP) which mediates degradation of tightly folded poly-

[1]Division of Immunology, Department of Biology, University of Konstanz, Konstanz, Germany; [2]Biotechnology Institute Thurgau at the University of Konstanz, Kreuzlingen, Switzerland

Correspondence: michael.basler@bitg.ch
*Franziska Oliveri and Steffen Johannes Keller contributed equally to this work and should be considered co-first authors

ubiquitylated proteins by making them accessible for the 19S regulator (Matyskiela et al, 2013; Gödderz et al, 2015; Schweitzer et al, 2016; Olszewski et al, 2019). Structural analysis revealed that the N-terminal heptapeptide of FAT10 is unstructured (Aichem et al, 2018) which might serve as a handle for the 19S particle, making VCP dispensable. Moreover, FAT10 is only loosely folded with a low melting temperature compared with ubiquitin (Wintrode et al, 1994; Aichem et al, 2018). Because IDPs are reported to be degraded by the 20S proteasome (Ben-Nissan & Sharon, 2014; Kumar Deshmukh et al, 2019), we investigated degradation of FAT10 by the 20S proteasome in vitro and in cellulo in this study.

# Results

### FAT10 is degraded by the 20S proteasome in vitro

Several in vitro studies showed that the 20S proteasome actively recognizes and degrades oxidized proteins and proteins with inherent structural disorders. Degradation by the 20S proteasome does not require ubiquitin tagging or the presence of the 19S RP and instead relies on recognition of unstructured regions of the proteins due to aging or oxidation, and of intrinsically unfolded proteins (summarized in the study by Ben-Nissan and Sharon [2014] and Kumar Deshmukh et al [2019]). Because nuclear magnetic resonance studies have shown that the ubiquitin-like modifier FAT10 contains an intrinsically disordered N-terminal tail (Aichem et al, 2018), FAT10 degradation by the 20S proteasome in the absence of the 19S RP was investigated.

Therefore, we used human 20S proteasome purified from LCL721.45 cells (Fig S1) and analyzed the degradation of recombinant FAT10 in vitro for 5 h. Untagged recombinant FAT10 is degraded by the 20S proteasome and, similar to experiments in cellulo (Schmidtke et al, 2019), FAT10 degradation was blocked by the proteasome inhibitor MG-132 (Fig 1A). These results were confirmed with Flag-tagged FAT10 purified from transiently transfected HEK293 cells (Fig 1B). In addition, the degradation of FAT10 conjugates by the 20S proteasome could be detected in this setting. Degradation of both, monomeric FAT10 and FAT10 conjugates, were inhibited by MG-132. Recent studies have shown that replacing the cysteine residues of FAT10 slows down the degradation of FAT10 monomer and conjugates in cellulo (Aichem et al, 2018). A similar stabilization of FAT10 was observed in this study in vitro (Fig 1C), which is in line with the previously raised hypothesis that the weak folding of FAT10 leads to fast proteasomal degradation as it was reported for IDPs (Aichem et al, 2018). Furthermore, it was reported that degradation of FAT10 is independent of VCP, whereas N-terminally truncated FAT10 was dependent on the unfolding activities of VCP (Aichem et al, 2018). Therefore, we compared the in vitro degradation by 20S proteasomes of WT FAT10 with FAT10-ΔAPNASC lacking the N-terminal disordered tail. Interestingly, degradation of FAT10-ΔAPNASC was almost completely arrested (Fig 1D), indicating that the disordered N-terminal tail is required for 20S-mediated degradation.

Taken together, our data show that FAT10 can be degraded by the 20S proteasome in vitro and suggests that this degradation is mediated by its characteristics as an intrinsically disordered protein.

### Treatment with *PSMC1* siRNA down-regulates mRNA and protein levels of Rpt2 and leads to accumulation of poly-ubiquitylated proteins

20S proteasome–mediated protein degradation has gained attention in the last years, but research in this field is limited by the difficulties of studying 20S proteasome degradation in cellulo (Kumar Deshmukh et al, 2019). To assess whether the results from our in vitro studies using purified 20S proteasomes can be confirmed in a physiological environment, a system to study the 20S proteasome in cellulo was established. An siRNA approach targeting the *PSMC1* gene which translates into Rpt2, one of the six AAA+ ATPases at the base of the 19S RP, was used to disturb 26S proteasome function. Because of the proposed key role of Rpt2 in the regulation of the 20S α-subunit gate opening and stabilization of the 26S complex (Köhler et al, 2001; Gillette et al, 2008), we hypothesized that knockdown of Rpt2 would impair the stability of the 19S RP and impair the 26S proteasome activity. We used a HEK293T cell line stably overexpressing Flag-FAT10 and transfected the cells with *PSMC1* siRNA. To increase the knockdown efficiency, transfection with siRNA was repeated on day 3. Rpt2 was down-regulated in *PSMC1* siRNA–treated cells, whereas Rpt2 levels stayed the same in unspecific control siRNA–treated cells (Fig 2A). Strikingly, poly-ubiquitylated proteins were accumulated on day 3, 5, and 6 post *PSMC1* siRNA transfection (Fig 2A), indicating the impairment of 26S proteasome function. This confirms the important role of Rpt2 in the degradation of substrate proteins by the 26S proteasome (Köhler et al, 2001). Analysis of 26S proteasomes in cell lysates by native gel electrophoresis and immunoblotting revealed a decreased amount of 26S proteasomes upon *PSMC1* knockdown (Fig 2B).

### siRNA-mediated knockdown of Rpt2 affects FAT10 degradation only mildly

This approach was used to investigate the degradation of FAT10 by the 20S proteasome in cellulo. Cells were transfected twice with siRNA, and a cycloheximide (CHX) chase experiment was performed on day 5 post first transfection. This time point was chosen because cells had the lowest Rpt2 levels and later on the viability of the cells was declining, most likely due to dysfunction of the 26S proteasome (data not shown). Rpt2 was down-regulated and poly-ubiquitylated proteins accumulated in *PSMC1* siRNA–treated cells (Fig 3A). Similar to in vitro analysis (Fig 1A and B), FAT10 was rapidly degraded in control siRNA–treated samples during 5 h of the CHX chase. In *PSMC1* siRNA—treated samples, an accumulation of FAT10 was already seen at time point 0 when starting the CHX chase, suggesting a role of the 26S proteasome in its degradation. However, immunoblotting and quantification of three independent experiments revealed a similar degradation kinetic during the CHX chase like the controls (Fig 3A and C). Nevertheless, degradation of poly-ubiquitylated proteins was markedly reduced in the *PSMC1* siRNA samples which show the dysfunction of the 26S proteasome (Fig 3B).

### miRNA-mediated knockdown of Rpt2 impairs the 26S proteasome

Even though the siRNA results suggest an impaired degradation of poly-ubiquitylated proteins in the absence of the 26S proteasome,

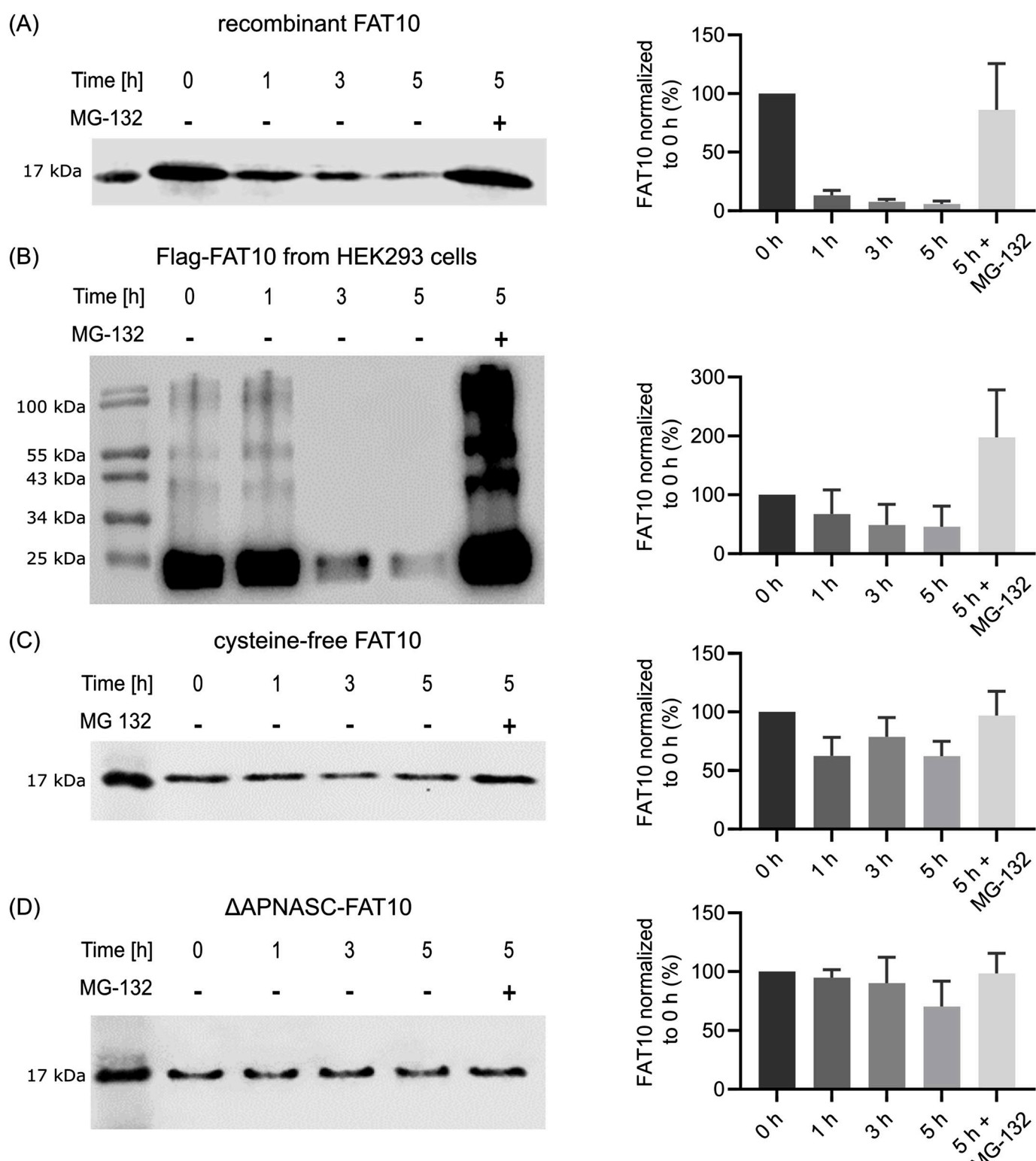

**Figure 1. FAT10 is degraded in vitro by purified 20S proteasome.**

300 ng purified 20S proteasome was incubated with different forms of FAT10 at 37°C for the indicated time in absence (−) or presence (+) of MG-132. Samples were analyzed by SDS–PAGE and Western blotting. Left panels show representative Western blots against FAT10 and right panels quantification of band intensities in percentage compared with time point 0. **(A)** Recombinant FAT10. **(B)** Flag-FAT10 isolated from transiently transfected HEK293 cells. **(C)** Stabilized cysteine-free FAT10. **(D)** ΔAPNASC-FAT10 lacking the N-terminal disordered tail. Results are shown as mean + SD. (A, C, D) n = 5, (B) n = 4.

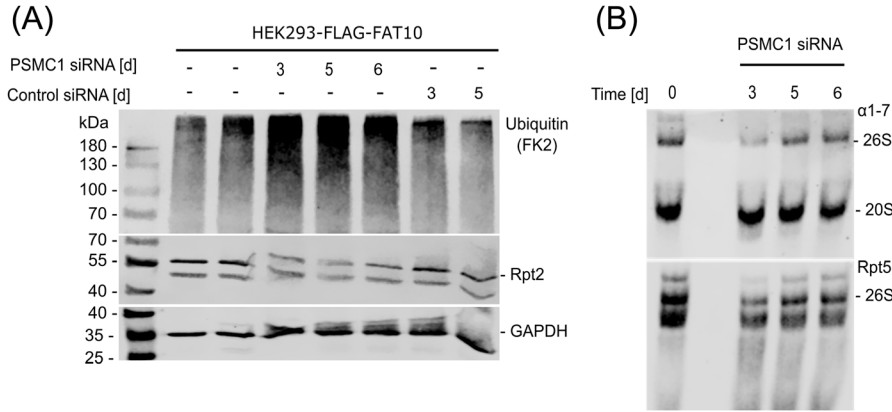

**Figure 2. Treatment with *PSMC1* siRNA down-regulates protein levels of Rpt2 and leads to accumulation of poly-ubiquitylated proteins.**
Cells were either transfected with *PSMC1* siRNA, scrambled control siRNA, or left untreated for the indicated time. A second transfection was performed on day 3. **(A)** Samples were lysed, and SDS–PAGE was performed followed by Western blotting against the indicated antibodies. GAPDH was used as loading control. **(B)** Cells were lysed and analyzed by native gel electrophoresis and Western blotting. An antibody directed against the alpha subunits 1–7 of the proteasome was used for detection of the 26S and 20S proteasome (upper panel). Rpt5 was used for specific detection of the 19S regulator which is part of the 26S proteasome (lower panel).

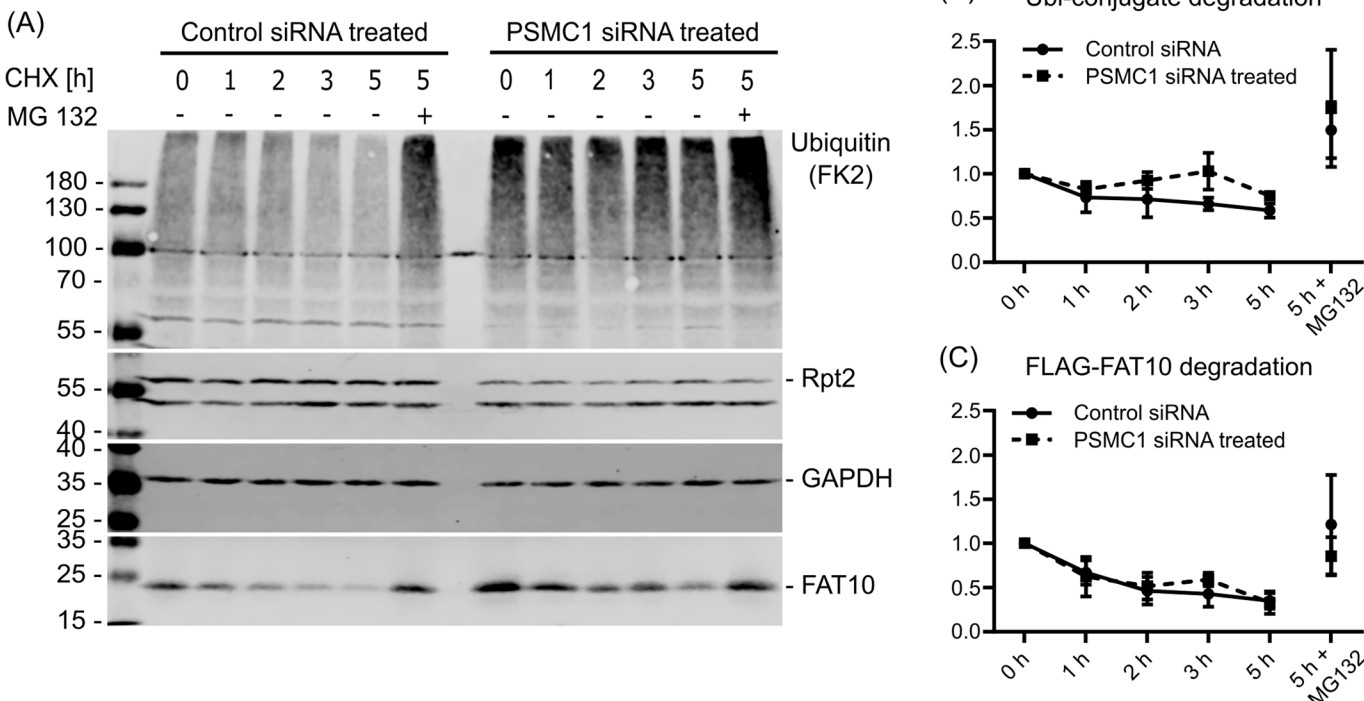

**Figure 3. FAT10 degradation upon *PSMC1* knockdown.**
Cells were either transfected with *PSMC1* siRNA or scrambled control siRNA on day 0 and day 3 and cycloheximide chase experiments were performed on day 5. Cells were collected at the indicated time points, and SDS–PAGE was performed followed by Western blotting with antibodies directed against the indicated proteins. GAPDH was used as loading control. **(A)** Western blots of one out of three independent experiments. **(B, C)** Degradation of (B) poly-ubiquitylated proteins and (C) FAT10 over time. Band intensities were quantified and normalized to GAPDH. All values are expressed as a mean ± SD, n = 3.

varying knockdown efficiency complicates data interpretation. To overcome this problem, a stable cell line inducibly expressing *PMSC1* miRNA was generated. The tetracycline responsive T-REx system (Thermo Fisher Scientific) was used. T-REx-293 cells were stably transfected with the plasmid pT-Rex-Dest30_PSMC1 encoding three different miRNAs directed against *PSMC1*. All three miRNA sequences are encoded on the same plasmid to ensure common integration, and the concerted GFP expression allowed to sort for positive cells and later to monitor stable expression of the introduced sequences. We found strong GFP

expression after induction with tetracycline (Fig 4A) and a strong *PSMC1* knockdown (Fig 4B). Time course analysis of the two different cell clones AU and AY revealed strong reduction of Rpt2 3 d post induction which goes along with an accumulation of poly-ubiquitylated proteins (Fig 4C), indicating the impairment of the 26S proteasome. A third clone (AK) did not show differences in accumulation of poly-ubiquitinated proteins and, therefore, was not used for further analyses. Furthermore, native gel analysis showed strongly decreased 26S expression 3 d post induction (Fig 4D). In contrast, the overall chymotrypsin-like activity of the

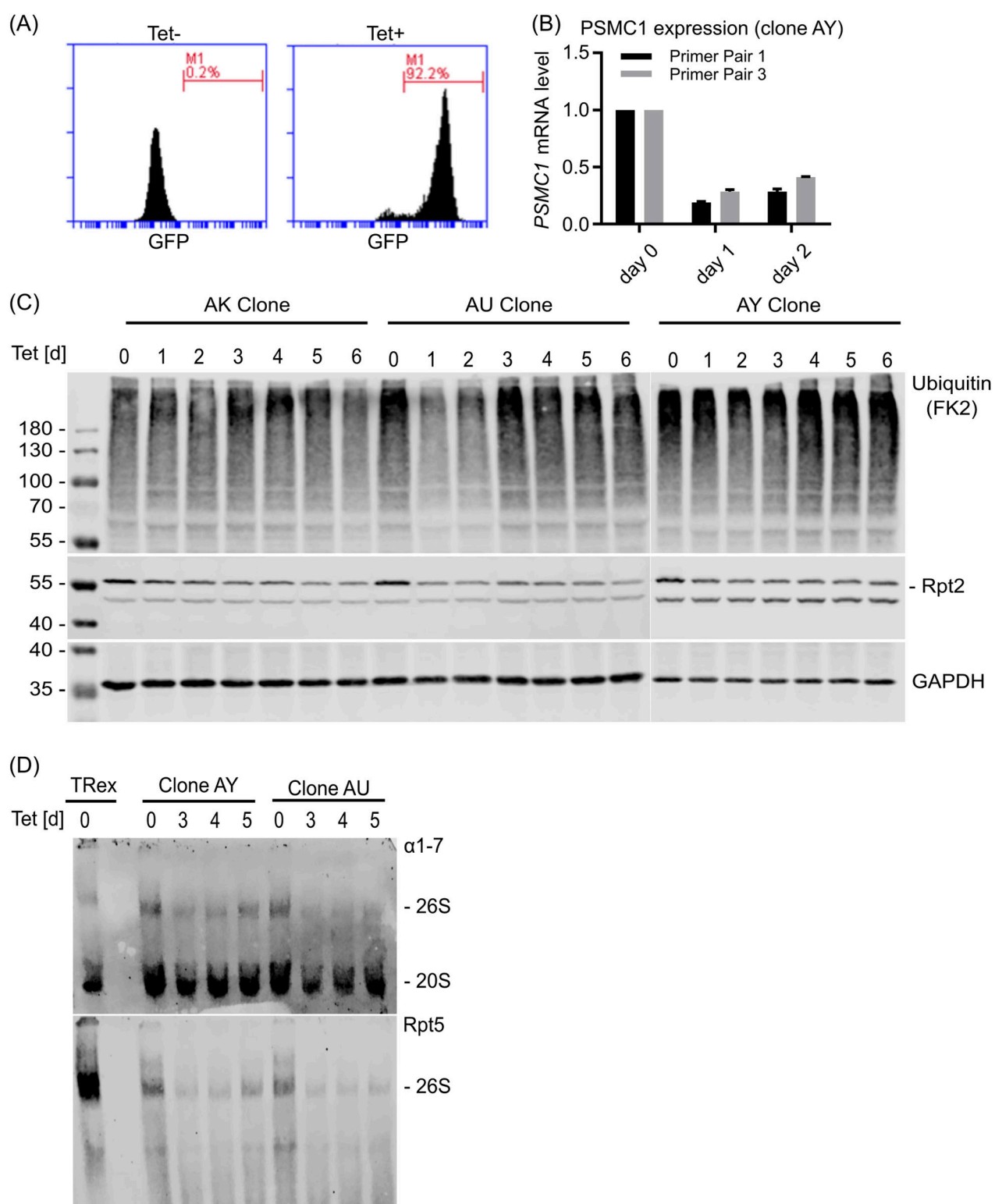

**Figure 4. miRNA-mediated knockdown of Rpt2 leads to accumulation of poly-ubiquitylated proteins and reduction of the 26S proteasomes.**
Tetracycline (Tet) responsive T-REx cells were stably transfected with a plasmid encoding for three miRNAs directed against *PSMC1* mRNA (encodes for the protein Rpt2). **(A)** The transfected plasmid concomitantly encodes for emGFP which allows monitoring protein expression upon tetracycline addition. Fluorescent signal of uninduced sample without tetracycline (left) and induced sample cultured with tetracycline for 5 d (right). **(B)** Efficient knockdown of *PSMC1* mRNA was confirmed by real-time RT-PCR. **(C)** Three single-cell clones were cultured with tetracycline for the indicated time, lysed, and analyzed by Western blotting for the expression of Rpt2 and ubiquitin conjugates. GAPDH served as a loading control. **(D)** Two single-cell clones were cultured with tetracycline for the indicated time, lysed, and analyzed by native gel

proteasome in cells was not impaired by *PSMC1* knockdown as assessed by hydrolysis of the cell-permeable fluorogenic substrate MeOSuc-GLF-AMC (Fig S2) and dose-dependent inhibition by bortezomib, indicating that the 20S proteasome is still functional. Taken together, these data strongly suggest an impaired function of the 26S proteasome due to dissociation of the complex in cells with decreased Rpt2 expression.

### miRNA-mediated knockdown of Rpt2 leads to FAT10 accumulation

Because we observed an efficient knockdown of Rpt2 and impaired 26S proteasome function, we performed FAT10 degradation kinetic experiments in these cells. To analyze the degradation of FAT10 independent of the 26S proteasome, we transfected the cells with Flag-FAT10 (pcDNA3.1-3xFLAG-His-FAT10) on day 3 post miRNA induction and performed a CHX chase experiment on day 5. Strong accumulation of poly-ubiquitylated proteins could be observed in *PSMC1* knockdown (Tet+) samples which was not markedly reduced during the chase, confirming the impairment of the 26S proteasome (Fig 5). In contrast to the in vitro data, FAT10 degradation was strongly impaired in these cells and showed a similar degradation kinetic as poly-ubiquitin conjugates.

Taken together, our data show that Rpt2 knockdown by miRNA can effectively impair the 26S proteasome, resulting in an accumulation of poly-ubiquitylated proteins. Hence, this system is an attractive tool to investigate in cellulo degradation of proteins by the 20S proteasome, overcoming the limitations of in vitro studies with isolated proteins. Furthermore, we could show that even though FAT10 is degraded by the 20S proteasome in vitro, this does not reflect FAT10 degradation in cellulo.

## Discussion

Ubiquitin- and 19S RP-independent degradation by the 20S proteasome has gained increasing attention during the last years. This degradation mechanism relies on the inherent structural disorder of proteins which can result, for example, from oxidation in oxidative stress situations (Aiken et al, 2011). Intrinsically disordered proteins were described to be susceptible to degradation by the 20S route because they do not require unfolding by the regulatory 19S cap (reviewed in the study by Ben-Nissan and Sharon [2014] and Kumar Deshmukh et al [2019]). Several proteins were identified as substrates for the 20S proteasome such as the tumor suppressors p53 (Camus et al, 2007), p73 (Asher et al, 2005), and retinoblastoma protein (Kalejta & Shenk, 2003; Sdek et al, 2005), the cell cycle regulators p27 (Tambyrajah et al, 2007) and p21 (Jin et al, 2003) and tau (David et al, 2002) and α-synuclein (Tofaris et al, 2001), both related to neurodegenerative diseases. Interestingly, all those proteasome substrates have very important functions in cell cycle regulation and cellular growth and, therefore, can be involved in

oncogenesis. Thus, tight regulation appears important and highlights the importance of 20S proteasome research.

Structural analysis of the ubiquitin-like modifier FAT10 by nuclear magnetic resonance studies revealed its weak folding and an intrinsically disordered N-terminal tail (Aichem et al, 2018). This suggests that FAT10 can be degraded by the 20S proteasome without the requirement of the 19S RP. Our data clearly show that recombinant and Flag-tagged FAT10, purified from transiently transfected cells, can be readily degraded by purified 20S proteasome in vitro. Interestingly, also FAT10 conjugates can be degraded in these in vitro digestion assays with purified 20S proteasomes, and the proteasome inhibitor MG-132 prevents the degradation of the FAT10 conjugates. This indicates that FAT10ylated substrates can be directly degraded by 20S proteasomes in vitro without the requirement of de-FAT10ylating enzymes. In contrast, a stabilized form of FAT10, in which all four cysteine residues are mutated, is degraded much slower by the 20S proteasome than the WT form. It was recently shown that mutating the cysteine residues maintains the general folding of FAT10 but leads to a stabilization of the normally flexible N-terminus (Aichem et al, 2018). In line with the previous results, degradation of FAT10-ΔAPNASC, in which the N-terminal tail is removed, was markedly slowed down, highlighting the important role of the disordered tail for degradation.

Taken together, the degradation of FAT10 in vitro by the 20S proteasome depends on its flexible N-terminal tail and its weak folding.

Even though our data clearly demonstrate FAT10 degradation in vitro, it was unclear whether this process actually occurs in cellulo. Studying the role of the 20S proteasome in cellulo is challenging because of the great abundance of the 26S proteasome and the importance of the ubiquitin–proteasome system in various essential cellular processes (Sherman & Goldberg, 2001). Past studies focusing on ubiquitin-independent degradation often used mutated forms of ubiquitin (reviewed in Hwang et al [2011]) to investigate degradation and were mostly conducted in vitro. FAT10 is degraded independently of ubiquitin and can itself target proteins for degradation via the 26S proteasome (Hipp et al, 2005; Schmidtke et al, 2009). To establish a system to study the 20S-dependent degradation of FAT10 in cellulo, we knocked down *PSMC1* by RNAi which translates into Rpt2, one of the six AAA+ ATPases at the base of the 19S RP. Those ATPases trigger a rotation of the α-subunits leading to opening of the gate of the 20S core particle and promotion of peptide substrate entry upon binding of ATP (Köhler et al, 2001; Gillette et al, 2008). We could show that transient siRNA and inducible miRNA-mediated knockdown leads to an accumulation of poly-ubiquitinated proteins which cannot be degraded in CHX chase experiments, indicating the impairment of the 26S proteasome. Furthermore, we could show a reduced proportion of 26S proteasomes in RNAi-treated samples by native gel analysis. Sahu et al recently reported a similar effect upon Rpn1 knockdown (Sahu et al, 2021), strengthening the principle of this

electrophoresis and immunoblotting. An antibody directed against the alpha subunits 1–7 of the proteasome was used for detection of the 26S and 20S proteasome (upper panel). Rpt5 was used for specific identification of the 19S regulator which is part of the 26S proteasome (lower panel). Untransfected cells served as a negative control.

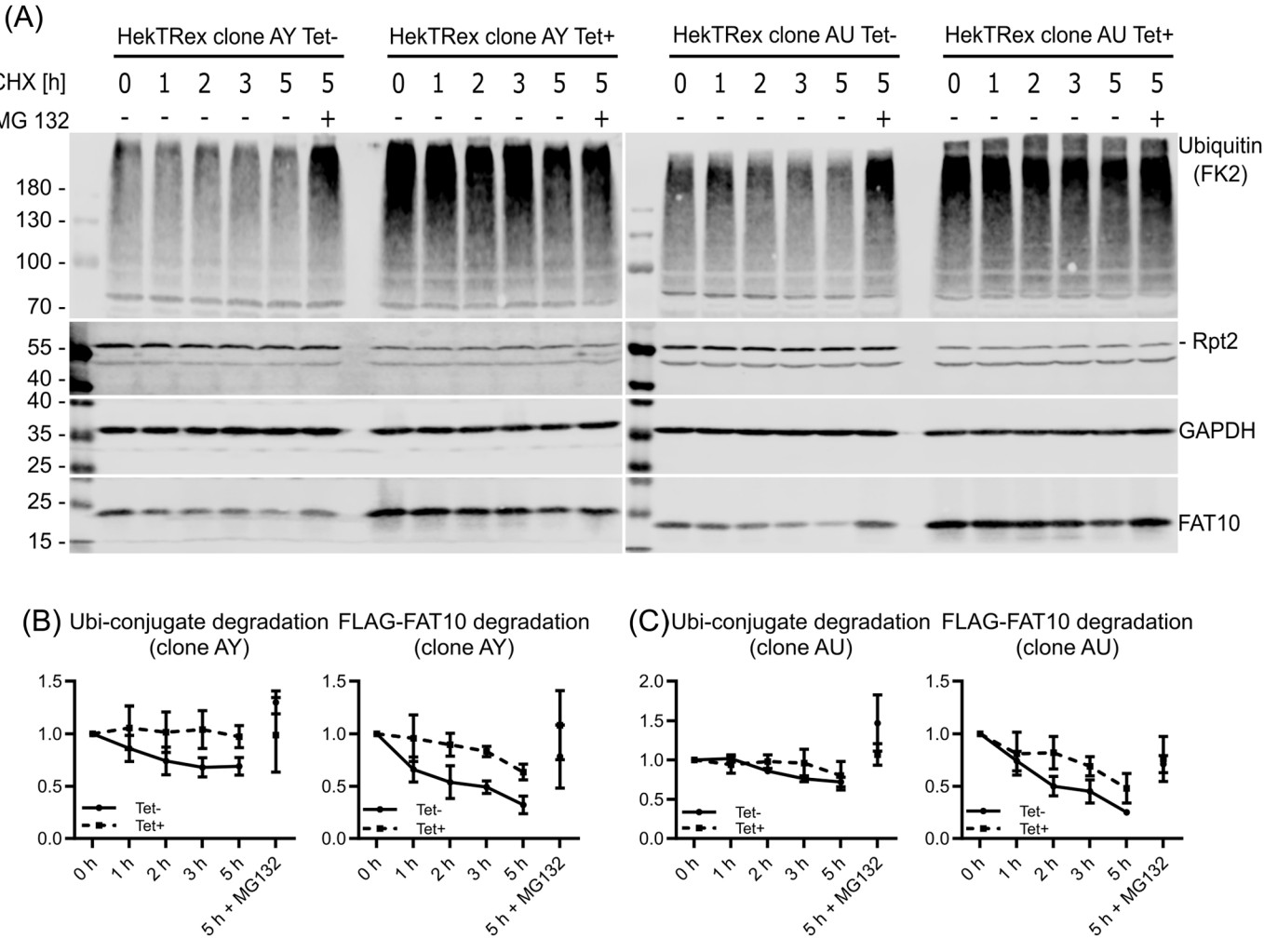

**Figure 5. FAT10 degradation depends on intact 26S proteasome function.**
Cycloheximide chase experiments were performed with cell clones AY and AU which were cultured with (+) or without (−) tetracycline (Tet) for 5 d. **(A)** Cells were collected at the indicated time points, and SDS–PAGE was performed followed by Western blotting with antibodies directed against the indicated proteins. GAPDH was used as loading control. **(B, C)** Quantification of degradation of (left) poly-ubiquitylated proteins and (right) FAT10 over time of the clones AY (B) and AU (C). Band intensities were quantified and normalized to GAPDH. All values are expressed as a mean ± SEM, n = 3 (independent experiments).

approach. Because impairment of the 26S proteasome affects many cellular functions, we could observe reduced cell growth after 6 d of knockdown (data not shown). Another study already reported difficulties to establish stable cell lines with knocked out 19S subunits (Tsvetkov et al, 2015), making our inducible system a great alternative for comparative studies of 26S proteasome-independent degradation in a controlled, in cellulo setting and will be of great interest to the field.

Poly-ubiquitin accumulation was present upon both si- and miRNA-mediated *PSMC1* knockdown. In siRNA-treated cells, FAT10 was accumulated after 5 d, indicating that in cellulo FAT10 degradation is at least partially dependent on the 26S proteasome. Degradation of FAT10 in CHX chase experiments seems not to be influenced by siRNA-mediated *PSMC1* knockdown, suggesting that FAT10 degradation is mediated by the 20S proteasome or that the residual 26S activity is sufficient to degrade FAT10 in cellulo. Because poly-ubiquitin conjugates are also partially degraded after

siRNA-mediated *PSMC1* knockdown, the latter seems reasonable. To obtain results that are more conclusive, an inducible miRNA-mediated knockdown of the *PSMC1* approach was chosen. Whereas FAT10 is largely degraded in non-induced cells, FAT10 is arrested to a similar extent as poly-ubiquitin conjugates when Rpt2 is down-regulated. These results strongly suggest that FAT10 degradation in cellulo is mediated by the 26S proteasome and that the contribution of the 20S alone is not measurable and insignificant within 5 h of degradation. This is in line with previous studies. In the yeast *S. cerevisiae* FAT10 degradation requires binding to Rpn10 of the 19S RP via the von Willebrand A domain. Degradation of FAT10 relies on Rpn10 in the 19S RP because it is accumulated when Rpn10 is knocked out (Rani et al, 2012). Furthermore, FAT10 degradation in cellulo is VCP independent but needs the unstructured N-terminal tail for this degradation. Together with our new data, we suggest that this unstructured region initiates degradation via the 26S proteasome. One hypothetical

mechanism could be that the unstructured region gets in contact with the ATPase ring of the 19S RP which initiates a pull into the complex as it was described for loosely folded proteins (Matyskiela et al, 2013; Schweitzer et al, 2016).

Our data further demonstrate the discrepancy in 20S-mediated protein degradation between in vitro and in cellulo processes. A possible explanation for this difference might be the concentrations of the different components used for in vitro degradation which do not reflect the situation in cells. Excess of FAT10 over the proteasome and the lack of putative competitors for degradation or binding to the 20S proteasome might enhance in vitro 20S-mediated FAT10 degradation. In addition, purified 20S proteasomes used for in vitro degradation might have different degradation properties compared with free 20S proteasomes located in the cell. For instance, α-rings might open up during purification and storage of 20S proteasomes, facilitating substrate entry into the 20S core. Furthermore, additional factors might control 20S-mediated degradation in cells, obviously absent in assays with purified 20S proteasomes. It is not yet clear how IDPs are targeted to the 20S proteasome for degradation and how they activate the proteolytic activity because degradation requires gate opening of the α-subunits (Sahu & Glickman, 2021). Previous data suggest interaction of certain features in IDPs with the α-ring which then promotes gating of the 20S proteasome and degradation of the proteins (Liu et al, 2003; Sahu & Glickman, 2021). Myers et al could demonstrate that not all IDPs can be degraded by the 20S proteasome and that a high degree of disorder is required for efficient degradation (Myers et al, 2018). Thus, FAT10 might still be too structured for efficient degradation in cellulo. This might be mediated by factors in cells keeping FAT10 in a conformation not allowing degradation in cellulo.

Taken together, in this study we established a new tool to investigate free 20S proteasome–mediated protein degradation in cellulo. Data obtained with this tool points out that data derived from in vitro 20S degradation studies cannot necessarily be translated to actual cellular processes in a physiological environment and require cautious data interpretation.

# Materials and Methods

## Cell culture

The human lymphoblastoid cell line LCL721.45 (DeMars et al, 1984) was cultured in IMDM supplemented with 10% FBS and 1% penicillin/streptomycin in a humidified 5% $CO_2$ atmosphere at 37°C. T-REx-293 cells, stably expressing the tetracycline repressor, were originally purchased from Thermo Fisher Scientific. The Flag-FAT10-overexpressing HEK293T cell line was previously described (Roverato et al, 2021). All HEK293-derived cell lines were cultured in DMEM supplemented with 10% FBS and 1% penicillin/streptomycin in a humidified 5% $CO_2$ atmosphere at 37°C. For T-REx-293 cells tetracycline negative serum from Bio&Sell GmbH was used. 0.05% trypsin–EDTA was used for detachment after washing with PBS. After centrifugation cell number was determined using an automated cell counter (Nexcelom), and cell density was adjusted. All cell culture reagents were purchased from Thermo Fisher Scientific if not stated otherwise.

## Cell transfection

Cells were grown in antibiotic-free medium (DMEM with 10% FBS) to a confluency of 80%. Plasmid DNA and TransIT-LT1 transfection reagent (Mirus Bio LLC) were mixed in a ratio of 1:3 (μg DNA:μl reagent) and added dropwise to the cells. Medium was replaced with fresh fully supplemented medium 24 h after transfection. For FAT10 degradation experiments, the plasmid pcDNA3.1-His-3xFLAG-Fat10, encoding FLAG-tagged FAT10 (Chiu et al, 2007), was used for transient transfection.

## Purification of Flag-FAT10 from transfected HEK293 cells

HEK293 cells were transfected with pcDNA6-His-3xFlag-FAT10 plasmid and TransIT-LT1 transfection reagent as described above. After 46 h, cells were harvested and lysed with PBS containing 0.5% NP40 and protease inhibitor mix (Roche) for 20 min on ice. Debris were removed by centrifugation (13,000g, 5 min). Nickel–IDA beads (Macherey-Nagel) were added to the supernatant and incubated at 4°C overnight. Samples were eluted from the beads with 300 mM imidazole (200 μl) in three fractions and filtered with an Amicon Ultra-15 3 K anion filter (Merck Millipore Ltd.).

## 20S proteasome and recombinant FAT10 purification

The 20S proteasome was purified as described previously (Groettrup et al, 1995; Schmidtke et al, 1996; Basler & Groettrup, 2012). In brief, cell pellets from LCL721.45 cells were collected and lysed with 100 mM KCl + 0.1% Triton X-100. The suspension was homogenized with a Dounce homogenizer and subsequently centrifuged for 30 min at 30,600g and 4°C. The supernatant was mixed with the anion exchanger DEAE Sephacel (GE Healthcare) and incubated overnight at 4°C while rotating. The next day, the DEAE mix was column purified and eluted with 500 mM KCl. $(NH_4)_2SO_4$ was added (35% of total volume), and samples were centrifuged twice for 20 min at 17,211g at 4°C. The resulting pellet was resolved in 100 mM KCl and incubated on ice for 60 min. Afterward, the dissolved pellet was added on a sucrose gradient (15–40%) and centrifuged for 16 h at 274,355g at 4°C.

After centrifugation, the sucrose gradient was separated into fractions of 700 μl and tested for proteasomal activity and further purified via a Resource Q column and fast protein liquid chromatography. Successful isolation of the 20S proteasome was confirmed with activity assay and SDS–PAGE.

Recombinant FAT10 was purified from transfected *Escherichia coli* BL21(DE3) CodonPlus cells (Stratagene) as described before using Ni-affinity chromatography and size-exclusion gel filtration (Aichem et al, 2018).

## In vitro degradation of FAT10

For in vitro degradation analysis, purified 20S proteasome was incubated with different FAT10 forms. 300 ng of purified 20S proteasome was added to 200 ng of recombinant FAT10 or 10 μl eluate (hFLAG-FAT10 from HEK293 cells). As a control for proteasomal-specific degradation, 10 μM proteasome inhibitor MG-132, starting 30 min before FAT10 addition, was used. Samples were incubated at

37°C for the indicated time in proteasome buffer (50 mM Tris, pH 7.5, 25 mM KCl, 10 mM NaCl, 1 mM MgCl$_2$, 1 mM DTT, 0.1 mM EDTA). After that, samples were boiled with 5× SDS sample buffer supplemented with 5% 2-mercaptoethanol at 95°C for 5 min. Samples were analyzed by SDS–PAGE and Western blotting.

## siRNA transfection

Cells were grown in a six-well dish to a confluency of 80% in antibiotic-free medium. *PSMC1* siRNA kit (sc-92427; Santa Cruz Biotechnology) was used according to the manufacturer's instructions to transfect Flag-FAT10-overexpressing Hek293T cells as described previously (Roverato et al, 2021). Transfection was repeated after 3 d to ensure prolonged knockdown. Unspecific scrambled siRNA (sc-37007; Santa Cruz Biotechnology) served as a control. Transfection was performed in FBS-free siRNA transfection medium (sc-36868; Santa Cruz Biotechnology). Medium was changed after 24 h of incubation.

## miRNA-mediated knockdown of PSMC1

For generation of a stable cell line with inducible knockdown of *PSMC1*, we used T-REx-293 cells and the BLOCK-iT Inducible Pol II miR RNAi Expression Vector Kit with EmGFP (Invitrogen) according to the manufacturer's instructions. In brief, we designed three different miRNAs targeting *PSMC1* mRNA using Life Technologies' RNAi Designer. The following oligonucleotides were cloned into pT-REx-DEST30 Gateway to generate the pT-REx-DEST30_PSMC1 Gateway plasmid: NM_002802.3_658_top (5′-TGCTGATAATATTC-AGGATGGGTGAGGTTTTGGCCACTGACTGACCTCACCCACTGAATATTAT-3′), NM_002802.3_658_bottom (5′-CCTGATAATATTCAGTGGGTGAGGTCA-GTCAGTGGCCAAAACCTCACCCATCCTGAATATTATC-3′), NM_002802.3_684_top (5′-TGCTGTTAGGAGGCTTTATACCCATCGTTTTGGCCACTGA-CTGACGATGGGTAAAGCCTCCTAA-3′), NM_002802.3_684_bottom (5′-CCTGTTAGGAGGCTTTACCCATCGTCAGTCAGTGGCCAAAACGATGGGTAT-AAAGCCTCCTAAC-3′), and NM_002802.3_711_top (5′-TGCTGCCAGGTGG-ACCATAGAGAATGGTTTTGGCCACTGACTGACCATTCTCTGGTCCACCTGG-3′), NM_002802.3_711_bottom (5′-CCTGCCAGGTGGACCAGAGAATGGTCAG-TCAGTGGCCAAAACCATTCTCTATGGTCCACCTGGC-3′). T-REx-293 cells were transfected with the pT-REx-DEST30_PSMC1 Gateway plasmid as described above. After 5 d, cells were treated with 10 µg/ml tetracycline for 6 h, and GFP-positive cells were single-cell sorted into 96-well plates to select for stably transfected single-cell clones. For single-cell sorting and subsequent GFP measurements, cells were harvested and resuspended in PBS containing 2% FBS and 2 mM EDTA. SYTOX Blue (Thermo Fisher Scientific) was included for live/dead cell discrimination. Single cell-sorting was performed with BD FACSAria III, and confirmatory GFP measurements were analyzed with BD Accuri C6 (both BD Biosciences). Generated cell lines were tested negative for mycoplasma.

## Real-time RT-PCR

To evaluate knockdown efficiency, RNA was isolated using the RNeasy Mini kit (QIAGEN) according to the manufacturer's instructions. 1 µg of RNA was reversely transcribed into cDNA using the cDNA Synthesis Kit (Biozym). Real-time RT-PCR was performed using the Blue S′Green qPCR Kit (Biozym) with three specific primers for *PSMC1* (PSMC1_F1 5′-AAGAAAAGCAAGAGGAGGAAAG-3′; PSMC1-R1 5′-CACAGATGTAGACACGATGG-3′; PSMC1_F2 5′-AGCAAACCAAACCT-CAGCC-3′; PSMC1_R2 5′-TCCCCTAGAATCAAATCCATCC-3′; PSMC1_F3 5′-ACACTACGTCAGCATTCTTTC-3′; PSMC1_R3 5′-ATCCATCAGCAC-CCCTATC-3′). Expression of *PSMC1* was normalized to the expression of *HPRT*, which served as housekeeping gene (hHPRT-fwd 5′-TGGA-CAGGACTGAACGTCTTG-3′; hHPRT-rev 5′-CCAGCAGGTCAGCAAAGAATTTA-3′). Results are shown as 2$^{-\Delta\Delta Ct}$ values.

## SDS–PAGE and Western blotting

Western blotting was performed as described previously (Oliveri et al, 2022). Shortly, cells were harvested and lysed in RIPA (50 mM Tris-buffered HCl, pH 7.5, containing 150 mM NaCl, 1% NP-40, 0.5% SDS) or NP40 lysis buffer (20 mM Tris [pH 7.6], 50 mM NaCl, 10 mM MgCl$_2$, 1% NP40 [Igepal]) supplemented with 1× protease inhibitor cocktail (Roche) for 30 min on ice. Cell debris was removed via centrifugation for 20 min at 18,000*g* and 4°C. Protein concentrations were determined using the Pierce BCA protein assay kit (Thermo Fisher Scientific), and equal amounts were subjected to SDS–PAGE. Separated proteins were transferred onto a 0.45-µm nitrocellulose membrane (Whatman Protran) by wet blotting at 110 V for 90 min or by semi-dry blotting using the Trans-Blot Turbo System (Bio-Rad). After transfer, membranes were rinsed with ddH$_2$O and blocked in Odyssey TBS Blocking Buffer (LI-COR Biosciences) for 60 min at room temperature. Membranes were incubated in primary antibody solutions overnight at 4°C (list of antibodies can be found in Table S1). Anti-GAPDH antibody served as loading control. IRDye800CW goat anti-rabbit or anti-mouse and IRDye680RD goat anti-mouse or anti-rabbit (LI-COR Biosciences) were used as secondary antibodies. Signals were analyzed with the LI-COR Odyssey Imager and the Image Studio Lite Version 5.2.

## CHX chase experiment

Cells were grown in six-well plates to a confluency of about 95%. Cells were treated with 50 µg/ml CHX (Merck) to stop protein synthesis. One sample was in addition treated with 10 µM proteasome inhibitor MG-132 (Sigma-Aldrich). Cells were harvested after 0, 1, 2, 3, and 5 h, and cell pellets were stored at –20°C. Samples were analyzed using SDS–PAGE followed by Western blotting.

## Native PAGE

Native PAGE was performed as described before (Elsasser et al, 2005; Njomen et al, 2018). In brief, cells were lysed in native lysis buffer (50 mM Tris [pH 8], 0.5 mM EDTA, 10% glycerol, and 4 mM ATP). Cell suspensions were subjected to seven cycles of freezing in liquid nitrogen and thawing at room temperature. BCA assay was performed to determine the protein concentration, and 30 µg were diluted with native lysis buffer to a final volume of 30 µl. Samples were carefully mixed with 7 µl of 5× sample buffer (250 mM Tris, pH 7.4; 50% glycerol, 60 ng/ml xylene cyanol) and loaded onto native 4% polyacrylamide gels (90 mM Tris base/boric acid [pH 8.3], 5 mM MgCl$_2$, 0.5 mM EDTA, 1 mM ATP, 0.1% APS, 0.1% temed). Electrophoresis was performed at 4°C using 1× Towbin's basic buffer,

starting at 50 V for 30 min followed by 110 V for 3 h. After incubation in SDS running buffer containing 2% SDS and 1.5% 2-mercaptoethanol for 15 min at RT, Western blotting was performed using the Trans-Blot Turbo System (Bio-Rad) for 20 min at 2.5 A. Primary antibodies anti-α1-7 (Table S1) and anti-Rpt5 (TBP1; Table S1) were used to visualize specifically the 20S or 19S/26S complexes, as described previously (Semren et al, 2015).

### Fluorogenic substrate assay

To investigate the activity of the 20S proteasome in *PSMC1* knockdown cells, the cell permeable substrate MeoSuc-GLF-AMC was used as previously described (Basler et al, 2018). Shortly, cells were treated with different bortezomib concentrations in PBS + 25 mM HEPES for 30 min at 37°C. The cell permeable substrate MeoSuc-GLF-AMC (Bachem) (in DMSO 10 mM) was added at 40 $\mu$M to the cells and incubated at 37°C. The fluorescence intensity in the wells containing the cells was measured at an excitation wavelength of 360 nm and emission wavelength of 465 nm (infinite M200 pro; TECAN).

### Statistical analysis

Quantification of band intensities of Western blots was performed using Image Studio Lite Version 5.2 (LI-COR Biosciences). Information about sample size is detailed in the figure legends. Statistical analyses were performed with GraphPad Prism Version 9 and are detailed in the figure legends.

## Data Availability

The authors will share all primary and analyzed data upon reasonable request. Please contact the corresponding author.

## Supplementary Information

## Acknowledgements

We thank Nicola Catone (Biotechnology Institute Thurgau at the University of Konstanz) for providing recombinant FAT10 and Annette Aichem for FAT10 expression constructs. This manuscript is dedicated to the late Marcus Groettrup to acknowledge his contributions to the initiation of the study. We thank the facility for flow cytometry (FlowKon) at the University of Konstanz for help with flow cytometry. This work was supported by the "Forschungspreis Walter Enggist" (to M Basler) and German Research Foundation Grant GR 1517/25-1 and SFB969 project C01.

### Author Contributions

F Oliveri: conceptualization, data curation, formal analysis, supervision, validation, investigation, and writing—original draft, review, and editing.

SJ Keller: data curation, formal analysis, investigation, and writing—review and editing.

H Goebel and GO Alvarez Salinas: data curation, formal analysis, and investigation.

M Basler: conceptualization, supervision, investigation, and writing—original draft, review, and editing.

### Conflict of Interest Statement

The authors declare that they have no conflict of interest.

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
