## [Reviewer comments · Life Science Alliance]

Life Science Alliance

The ubiquitin-like modifier FAT10 is degraded by the 20S proteasome in vitro but not in cellulose

Franziska Oliveri, Steffen Keller, Heike Goebel, Gerardo Alvarez Salinas, and Michael Basler

DOI: <https://doi.org/10.26508/lsa.202201760>

Corresponding author(s): Michael Basler, University of Konstanz

Review Timeline:

Submission Date:	2022-10-07
Editorial Decision:	2022-10-31
Revision Received:	2023-02-28
Editorial Decision:	2023-03-21
Revision Received:	2023-03-23
Accepted:	2023-03-23

Transaction Report:

October 31, 2022

Re: Life Science Alliance manuscript #LSA-2022-01760

Dr. Michael Basler
University of Konstanz
Department of Biology, Division of Immunology
Universitätsstrasse 10
Konstanz 78457
Germany

Dear Dr. Basler,

Thank you for submitting your manuscript entitled "The ubiquitin-like modifier FAT10 is degraded by the 20S proteasome in vitro but not in cellulose" to Life Science Alliance. The manuscript was assessed by expert reviewers, whose comments are appended to this letter. We invite you to submit a revised manuscript addressing the Reviewer comments.

Thank you for this interesting contribution to Life Science Alliance. We are looking forward to receiving your revised manuscript.

Sincerely,

B. MANUSCRIPT ORGANIZATION AND FORMATTING:

Reviewer #1 (Comments to the Authors (Required)):

In this manuscript the authors describe a system in which the function of the 26S proteasome can be disrupted by RNAi of the 19S regulatory subunit Rpt2. The authors demonstrate that the ubiquitin-like modifier FAT-10 is degraded in vitro by the 20S proteasome, but that its degradation in cells is retarded when the function of the 26S proteasome is impaired. This implies that in vitro studies on proteasome-catalyzed degradation using purified proteins may not reflect the process in intact cells. The manuscript is clearly written and the experiments presented are well-defined. The system described may be of value to others who wish to study the function of the 26S proteasome in intact cells. However additional studies are required to more firmly establish the conclusions drawn by the authors.

As the authors state, the 26S proteasome which is involved in virtually all cell processes, when impaired produced a decline in cell viability after 5-6 days. Therefore, one can speculate that the observed effect on the degradation of FAT-10 following 26S impairment may be due to something other than direct catalysis of FAT-10 by the 26S proteasome. For example, impairment of the 26S proteasome may lead to accumulation of an endogenous inhibitor of the 20S proteasome, and that this inhibitor is responsible for the slowed FAT-10 degradation. The authors even state in their discussion that additional factors might control 20S proteasome-mediated degradation in cells. Therefore, it is essential that the authors determine whether the 20S proteasome is totally viable in the cells in which the 26S proteasome is impaired.

Some other points that need clarification:

Fig.1D After 5h incubation, APNASC-FAT-10 displays 2 bands and the band intensities which are decreased relative to control, suggest increased degradation. Please explain.

Fig. 4D Is there a down-regulation of the 20S proteasome after 3h?

Reviewer #2 (Comments to the Authors (Required)):

The authors use FAT10 to address whether in vitro degradation data observed with purified 20S proteasomes reflect the situation in living cells. They show that FAT10 is degraded in vitro by 20S proteasomes, and the degradation depends on the presence and exposure of an N-terminal unstructured domain. Knockdown leading to a depletion of the 19S regulatory particle (RP)ATPase subunit Rpt2 in human cells is shown to result in a reduction of 19S RP and accumulation of ubiquitylated proteins, at least in some of the experiments (see below). At the same time, the degradation of FAT10 is inhibited. These findings together lead the authors to the conclusions that the results obtained with in vitro degradation assays do not necessarily reflect the situation in vivo and that "that FAT10-degradation in cellulo is mediated by the 26S proteasome, and that the contribution of the 20S alone is not measurable and insignificant within 5 h of degradation".

Overall, this is a both interesting and valuable study addressing the critical issue of the function of 20S proteasome in vivo. There are a few small items that, in my opinion, deserve further attention and clarification.

It is not clear how the authors ascertain that their 20S proteasome preparation is devoid of any contamination with 19S particles. The Coomassie stain shown in figure S1 is consistent with a possibility for such contamination. Was addition of ATP-gamma-S or an ATP-depleting system tested during the degradation assays to exclude contribution of 19S particle contribution? How efficient would the 26S proteasome be in such in vitro experiments (see also the last point).

In figure S2 B, it is unclear which of the primer pairs (or a mixture) shown in figure A is used. From the western blot, it is not clear how effective the knockdown is at the protein level, as the bands that the authors identify as "non-specific" and the GAPDH bands so some extent change in the same way as the Rpt2 band, maybe with the exception of the bands in the 45 and 66 hour lanes. This is probably in part due to uneven loading. The authors interpret the data as a "rather inefficient knockdown of Rpt2 at the protein level", although this is difficult to judge as described above. By contrast, as suggested by the blot shown in Figure 3A, the Rpt2 knockdown appears to be effective also at the level of the protein in a modified version of the experiment. Effective Rpt2 knockdown is also shown for the stable integration of siRNA constructs (figure 4C). The blots shown in figures S2 and 2A

are not convincing in the same way, as discussed to some extent by the authors. It seems as if this is mainly due to a loading issue in Fig S2, and an electrophoresis or blotting issue in Figure 2A. Therefore, it is questionable whether these figures are helpful. I suggest to either reload samples from these experiments, or omit these figures. The quality of blots in figures 3A and 4C is superior (without blotting or loading issues) to the above-mentioned ones.

The phrase "...were stably transfected with the plasmid pT-Rex-Dest30_PSMC1 encoding three different miRNAs directed against PSMC1." could be misunderstood. At least if my understanding is correct, three different siRNAs were tested separately. If true, I suggest to modify the phrase e.g. as follows: "...were stably transfected with the three different plasmid pT-Rex-Dest30_PSMC1 encoding distinct miRNAs directed against PSMC1."

Even though I share the interpretation of the authors that it is likely a reduction in 26S proteasomes that is causing FAT10 stabilization, I think it is important to ascertain that the miRNA treatment does not, in some way, indirectly affect the 20S core particle. The reason why I bring this up is figure 4B, wherein it seems as if also the levels of the 20S CP are reduced after tetracycline-induced expression of siRNAs targeting PSMC1/RPT2. This might again be a loading issue, but deserves clarification.

There is a bit of an inconsistency between the in vivo experiments shown using different knockdown strategies. While transient PSMC1 knockdown experiments shown in figure 2A lead to a strong accumulation of ubiquitylated proteins over time, no clear such accumulation is observed with the cell lines stably transfected with constructs mediating tetracycline-induced PSMC1 knockdown. Nonetheless, the authors conclude: "Time course analysis of two different cell clones revealed strong reduction of Rpt2 three days post induction which goes along with an accumulation of poly-ubiquitylated proteins (Figure 4C),...". Instead, clone AU shows first a decline in signals for ubiquitylated signals. After 3h, the signals look similar to those at 0h. Which of the two different clones shown do the authors refer to in their statement? Is "AK" a control cell line?

There remains the possibility that FAT10 can be degraded, also in vivo, by both the 20S and the 26S forms of the proteasome, with the latter being more efficient. This could explain a lack of a clear stabilization in the cycloheximide chase experiment, where the proteasome does not have to deal with a constant de novo supply of FAT10 and other unstable proteins from ongoing translation. This would fit to a possibility raised by the authors when they discuss that ratios of substrates and proteasomes may be a reason for the apparent discrepancies between in vivo and in vitro data. If so, the expression levels of FAT10 could be a critical parameter in the in vivo experiments. If the 26S proteasome is more efficient in FAT10 degradation than the 20S proteasome, one would expect this to be detectable also in vitro, unless the 20S proteasome is artificially activated, as the authors reasonably argue.

Reviewer #3 (Comments to the Authors (Required)):

Oliveri and co-workers compared the ability of the 20S proteasomes to degrade FAT10 in vitro with in cellulo. This is a relevant question as far too often in vitro data with recombinant proteins are directly extrapolated to the physiological conditions in cells. Unfortunately, the data in the paper are not of particular high quality with the risk of being overinterpreted. Controls are missing while at the same time the study elaborates on less important points. Also spending an entire figure and section on generating an inducible cell line for depletion of a 19S component seems disproportional to the actual data that address the main question.

Comments

The western blots are generally of poor quality (heavily pixelated, disturbed bands without clear separation between the lanes). Moreover, measuring band intensities in a western blot is a semiquantitative method as one cannot be sure that there is a linear correlation between band intensity and protein levels.

Fig. 1B. Are the conjugates proteins modified by FAT10 or could it also be FAT10 being modified by, for example, ubiquitin. It looks like these conjugates are only observed with FLAG-tagged FAT10, which may be a concern as the FLAG tag contains two lysine residues that may be modified.

In Fig 1D, there is the appearance of a truncated fragment of the mutant FAT10 after 5 hours in the absence or presence of proteasome inhibitor. Is this due to residual proteasome activity that trims the protein or caused by another protease that is co-purified with the recombinant FAT10 or proteasomes? Does this band also appear if the recombinant FAT10 is incubated for 5 hours in the absence of proteasomes? Are both bands included in the quantification of the western blot or only the band corresponding to the full-length mutant FAT10?

In the text and figures the authors refer to the disappearance of ubiquitin conjugates as "ubiquitin degradation". Unlike FAT10, ubiquitin is typically not being degraded by the proteasome but is recycled. The authors may be referring here to degradation of proteins that are modified with ubiquitin instead of ubiquitin itself. However, the disappearance of ubiquitin conjugates cannot be used as a proxy for degradation of ubiquitylated proteins as their disappearance can also be due to deubiquitylation of proteins. Instead, the authors can address this by analyzing the levels of a substrate for ubiquitin-dependent proteasomal degradation.

Fig 3. It is difficult to draw conclusions from this experiment. The effect on ubiquitin conjugates in the quantification is modest. Having this as a positive control, it is hard to say whether there is an effect on FAT10 turnover. If the authors on top of that argue in the discussion that the discrepancy between the data with siRNA transfection in Fig 3 and the inducible depletion in Fig 5 are due to the conditions with the siRNA being suboptimal I see little reason for including those experiments in the manuscript.

Fig 4D. Why is the control AK cell line not included in this experiment?

The authors write "... FAT10ylated substrates can be directly grabbed by 20S proteasomes in vitro...". This anthropomorphism may be misplaced as the 20S proteasomes lacks subunits that actively engage in protein recruitment.

Reviewer #1 (Comments to the Authors (Required)):

In this manuscript the authors describe a system in which the function of the 26S proteasome can be disrupted by RNAi of the 19S regulatory subunit Rpt2. The authors demonstrate that the ubiquitin-like modifier FAT-10 is degraded in vitro by the 20S proteasome, but that its degradation in cells is retarded when the function of the 26S proteasome is impaired. This implies that in vitro studies on proteasome-catalyzed degradation using purified proteins may not reflect the process in intact cells. The manuscript is clearly written and the experiments presented are well-defined. The system described may be of value to others who wish to study the function of the 26S proteasome in intact cells. However additional studies are required to more firmly establish the conclusions drawn by the authors.

As the authors state, the 26S proteasome which is involved in virtually all cell processes, when impaired produced a decline in cell viability after 5-6 days. Therefore, one can speculate that the observed effect on the degradation of FAT-10 following 26S impairment may be due to something other than direct catalysis of FAT-10 by the 26S proteasome. For example, impairment of the 26S proteasome may lead to accumulation of an endogenous inhibitor of the 20S proteasome, and that this inhibitor is responsible for the slowed FAT-10 degradation. The authors even state in their discussion that additional factors might control 20S proteasome-mediated degradation in cells. Therefore, it is essential that the authors determine whether the 20S proteasome is totally viable in the cells in which the 26S proteasome is impaired.

Reply: The day of analysis was chosen to be day 5 for the degradation experiments because at this time point we did not observe changes in the viability, which was only observed 1-2 days later. However, in order to address the reviewers concern, we performed fluorescence-based activity assays to determine the functionality of the 20S proteasome. This assay uses the cell-permeable substrate MeO-Suc-GLF-AMC to measure chymotrypsin-like degradation. This short fluorogenic peptide can be degraded by the 20S proteasome independent of the 19S regulator in cells. To control for proteasome-dependent degradation of the substrate we used the pan proteasome inhibitor bortezomib, which reduced the activity in a dose-dependent manner indicating proteasome dependent MeO-Suc-GLF-AMC degradation. The results of three clones on day 3 and 5 of miRNA-induction are shown in the supporting information as figure S2 and demonstrate equal activity of *PSMC1* knockdown and control cells. Similar results were obtained with the proteasome inhibitor MG-132 (not shown). This data shows that the function of the 20S proteasome core particle is not disturbed in cells with impaired 26S function.

Some other points that need clarification:

Fig.1D After 5h incubation, Δ APNASC-FAT-10 displays 2 bands and the band intensities which are decreased relative to control, suggest increased degradation. Please explain.

Reply: We agree that in this Western Blot at time point 5 hours we see a partial degradation of the Δ APNASC-FAT10. However, compared to recombinant FAT10 (Fig. 1A), in which one can see an almost complete degradation after 1 hour, the degradation of the Δ APNASC-FAT-10 is strongly delayed. Furthermore, in other experiments, degradation at 5 hours could barely be observed. Additionally, the quantification of 4 different independent experiments did not show a significant degradation after 5 hours. Nevertheless, to avoid this visual discrepancy after 5 hours we replaced the Western Blot with a more representative example.

Fig. 4D Is there a down-regulation of the 20S proteasome after 3h?

Reply: Figure 4D shows the analysis of 26S and 20S proteasome 3 days (not hours) post induction. We agree that the native gel might suggest a reduction of 20S proteasome on day 3 and therefore performed the activity assay as described in response to question 1. We could not detect any differences in proteasome activity on day 3 or 5, indicating that there was no reduction of the 20S proteasome as the native gel might suggest. Since quantification of native gels is often rather difficult, we think that the weaker band rather results from loading differences but this cannot be determined since this analysis does not allow the measurement of a loading control.

Reviewer #2 (Comments to the Authors (Required)):

The authors use FAT10 to address whether in vitro degradation data observed with purified 20S proteasomes reflect the situation in living cells. They show that FAT10 is degraded in vitro by 20S proteasomes, and the degradation depends on the presence and exposure of an N-terminal unstructured domain. Knockdown leading to a depletion of the 19S regulatory particle (RP)ATPase subunit Rpt2 in human cells is shown to result in a reduction of 19S RP and accumulation of ubiquitylated proteins, at least in some of the experiments (see below). At the same time, the degradation of FAT10 is inhibited. These findings together lead the authors to the conclusions that the results obtained with in vitro degradation assays do not necessarily reflect the situation in vivo and that "that FAT10-degradation in cellulo is mediated by the 26S proteasome, and that the contribution of the 20S alone is not measurable and insignificant within 5 h of degradation". Overall, this is a both interesting and valuable study addressing the critical issue of the function of 20S proteasome in vivo. There are a few small items that, in my opinion, deserve further attention and clarification.

It is not clear how the authors ascertain that their 20S proteasome preparation is devoid of any contamination with 19S particles. The Coomassie stain shown in figure S1 is consistent with a possibility for such contamination. Was addition of ATP-gamma-S or an ATP-depleting system tested during the degradation assays to exclude contribution of 19S particle contribution? How efficient would the 26S proteasome be in such in vitro experiments (see also the last point).

Reply: Our protocol used for isolation of the 20S proteasome involves a purification step via FPLC. Due to the different size of the 26S proteasome, both types of proteasomes elute in different fractions. Hence, the 26S proteasome should not be isolated along with the 20S proteasome. To experimentally confirm this, we analyzed the presence of the 19S subunits Rpt1, Rpt2, Rpt5, and Rpt6 in isolated 20S vs. 26S proteasome samples by immunoblotting. In contrast to the 26S proteasome preparation, none of the investigated 19S subunits could be detected in the 20S preparation. The alpha subunit iota was present in both 20S and 26S proteasome preparations. These data show that the 20S proteasome preparation is devoid of any 19S particles. We included the data in the supporting information in figure S1B. Therefore, ATP-gamma-S or ATP-depleting systems were not used.

In figure S2 B, it is unclear which of the primer pairs (or a mixture) shown in figure A is used.

Reply: Please excuse the lack for clarity of the labels in figure S2. The "primer pairs" just refer to different primers tested for real-time RT-PCR as indicated in the figure legend. The used siRNA mixture was the same for all experiments. As suggested in the comment below, we decided to omit figure S2 to avoid confusion of the different setups.

From the western blot, it is not clear how effective the knockdown is at the protein level, as the bands that the authors identify as "non-specific" and the GAPDH bands to some extent change in the same way as the Rpt2 band, maybe with the exception of the bands in the 45 and 66 hour lanes. This is probably in part due to uneven loading. The authors interpret the data as a "rather inefficient knockdown of Rpt2 at the protein level", although this is difficult to judge as described above. By contrast, as suggested by the blot shown in Figure 3A, the Rpt2 knockdown appears to be effective also at the level of the protein in a modified version of the experiment. Effective Rpt2 knockdown is also shown for the stable integration of siRNA constructs (figure 4C). The blots shown in figures S2 and 2A are not convincing in the same way, as discussed to some extent by the authors. It seems as if this is mainly due to a loading issue in Fig S2, and an electrophoresis or blotting issue in Figure 2A. Therefore, it is questionable whether these figures are helpful. I suggest to either reload samples from these experiments, or omit these figures. The quality of blots in figures 3A and 4C is superior (without blotting or loading issues) to the above-mentioned ones.

Reply: Please note that the experimental setup in figure S2 is different from the one in figure 3. Initially, we transfected the cells only once with siRNA which did not result in markedly reduced Rpt2 levels as shown in Figure S2B. Therefore, we decided to repeat the transfection with siRNA on day 3 which is then shown in Figure 2A. As suggested by the reviewer, we decided to omit the previous supplementary figure S2 and delete the respective paragraph in the results section to prevent confusion.

The phrase "...were stably transfected with the plasmid pT-Rex-Dest30_PSMC1 encoding three different miRNAs directed against PSMC1." could be misunderstood. At least if my understanding is correct, three different siRNAs were tested separately. If true, I suggest to modify the phrase e.g. as follows: "...were stably transfected with the three different plasmid pT-Rex-Dest30_PSMC1 encoding distinct miRNAs directed against PSMC1."

Reply: Indeed, all three miRNAs are encoded on the same plasmid to ensure that all three of them are integrated together upon transduction. This procedure was suggested by the manufacturer (Thermo Fisher) to increase knockdown efficiency. We included a respective sentence in the methods section to make this clear.

Even though I share the interpretation of the authors that it is likely a reduction in 26S proteasomes that is causing FAT10 stabilization, I think it is important to ascertain that the miRNA treatment does not, in some way, indirectly affect the 20S core particle. The reason why I bring this up is figure 4B, wherein it seems as if also the levels of the 20S CP are reduced after tetracycline-induced expression of siRNAs targeting PSMC1/RPT2. This might again be a loading issue, but deserves clarification.

Reply: In order to address the reviewers concern, we performed fluorescence-based activity assays to determine the functionality of the 20S proteasome. This assay uses the cell-permeable substrate MeO-Suc-GLF-AMC to measure chymotrypsin-like degradation. This short fluorogenic peptide can be degraded by the 20S proteasome independent of the 19S regulator. To control for proteasome-dependent degradation of the substrate we used the pan proteasome inhibitor bortezomib, which reduced the activity in a dose-dependent manner indicating proteasome dependent MeO-Suc-GLF-AMC degradation. The results of three clones on day 3 and 5 of miRNA-induction are shown in the supporting information as figure S2 in the revised manuscript and demonstrate equal activity of PSMC1 knockdown and control cells. Similar results were obtained with the proteasome inhibitor MG-132 (not shown). This data show that the function of the 20S proteasome core particle is not disturbed in cells with impaired 26S function.

There is a bit of an inconsistency between the in vivo experiments shown using different knockdown strategies. While transient PSMC1 knockdown experiments shown in figure 2A lead to a strong accumulation of ubiquitylated proteins over time, no clear such accumulation is observed with the cell lines stably transfected with constructs mediating tetracycline-induced PSMC1 knockdown. Nonetheless, the authors conclude: "Time course analysis of two different cell clones revealed strong reduction of Rpt2 three days post induction which goes along with an accumulation of poly-ubiquitylated proteins (Figure 4C),...". Instead, clone AU shows first a decline in signals for ubiquitylated signals. After 3h, the signals look similar to those at 0h. Which of the two different clones shown do the authors refer to in their statement? Is "AK" a control cell line?

Reply: We agree that the reduction of poly-ubiquitinated proteins on day 1 and 2 appears contra intuitive. Indeed, we did not expect an accumulation during the first few days as the 19S regulator is quite stable in the cells with a turn-over rate of about two days. Since already generated Rpt2 will still be present in the assembled complexes in induced cells, the knockdown will appear to functionally impair protein degradation once the pre-existing 26S proteasomes disappear over time. Therefore, we decided to perform the FAT10 degradation experiments on day 5.

In the AK cell line, Rpt2 knockdown was not that efficient. Consistently, poly-ubiquitin accumulation was not as prominent as in the AU and AY clone. Therefore, clone AK was not used in further experiments. We mention this in the revised manuscript.

We apologize for mistakenly inserting the wrong blot for clone AY in figure 4C. This blot was replaced with the correct blot in the revised manuscript. The correct blot shows a clear accumulation of poly-ubiquitinated protein after induction. It seems that the observed accumulation of poly-ubiquitinated proteins after Rpt2 knockdown induction is sufficient to delay FAT10 degradation (Figure 5B). Furthermore, Figure 5A (at 0 hours) shows a strong difference in ubiquitin accumulation between non-induced and induced clones AU and AY 5 days post induction.

There remains the possibility that FAT10 can be degraded, also in vivo, by both the 20S and the 26S forms of the proteasome, with the latter being more efficient. This could explain a lack of a clear stabilization in the cycloheximide chase experiment, where the proteasome does not have to deal with a constant de novo supply of FAT10 and other unstable proteins from ongoing translation. This would fit to a possibility raised by the authors when they discuss that ratios of substrates and proteasomes may be a reason for the apparent discrepancies between in vivo and in vitro data. If so, the expression levels of FAT10 could be a critical parameter in the in vivo experiments. If the 26S proteasome is more efficient in FAT10 degradation than the 20S proteasome, one would expect this to be detectable also in vitro, unless the 20S proteasome is artificially activated, as the authors reasonably argue.

Reviewer #3 (Comments to the Authors (Required)):

Oliveri and co-workers compared the ability of the 20S proteasomes to degrade FAT10 in vitro with in cellulo. This is a relevant question as far too often in vitro data with recombinant proteins are directly extrapolated to the physiological conditions in cells. Unfortunately, the data in the paper are not of particular high quality with the risk of being overinterpreted. Controls are missing while at the same time the study elaborates on less important points. Also spending an entire figure and section on generating an inducible cell line for depletion of a 19S component seems disproportional to the actual data that address the main question.

Comments

The western blots are generally of poor quality (heavily pixelated, disturbed bands without clear separation between the lanes). Moreover, measuring band intensities in a western blot is a semiquantitative method as one cannot be sure that there is a linear correlation between band intensity and protein levels.

Reply: We apologize for the poor quality of western blots, which must have been introduced in PDF generation. We will provide high resolution pictures in the revision. In our opinion, the bands are clearly visible except for GAPDH in figure 5. Unfortunately, high amounts of proteins have to be loaded to detect FAT10. Additionally, to show all samples on the same blot, 15-well combs have to be used, which lead to poorly separated GAPDH bands between lanes.

In our Western Blots we used the LI-COR Odyssey system to detect proteins, which allows quantitative relative comparisons between different samples. LI-COR Odyssey is a flexible, IR laser-based instrument. The platform facilitates protein quantification in a wide linear range. LI-COR Odyssey near-IR protein quantification discern strong and weak bands on a western blot accurately because fluorescent detection is static, unlike the dynamic light generation in chemiluminescence. Hence, this system allows an accurate quantification of bands.

Fig. 1B. Are the conjugates proteins modified by FAT10 or could it also be FAT10 being modified by, for example, ubiquitin. It looks like these conjugates are only observed with FLAG-tagged FAT10, which may be a concern as the FLAG tag contains two lysine residues that may be modified.

Reply: Apart from the FLAG tag, FAT10 also possesses internal lysines which could potentially be modified. However, since putative FAT10 conjugates can be detected up to 130 kDa (which would correspond to more than 15 ubiquitins) it seems rather unlikely that the observed bands derive from FAT10 being modified by ubiquitin.

In Fig 1D, there is the appearance of a truncated fragment of the mutant FAT10 after 5 hours in the absence or presence of proteasome inhibitor. Is this due to residual proteasome activity that trims the protein or caused by another protease that is co-purified with the recombinant FAT10 or proteasomes? Does this band also appear if the recombinant FAT10 is incubated for 5 hours in the absence of proteasomes? Are both bands included in the quantification of the western blot or only the band corresponding to the full-length mutant FAT10?

Reply: We agree that in this Western Blot at time point 5 hours we see a partial degradation of the Δ APNASC-FAT10. This second band has only been observed in one out of four experiments. Why such a second band has been observed is not known. Both bands have been used for quantification in this experiment. To avoid confusion, the image was replaced by a representative Western Blot. Importantly, compared to recombinant FAT10 (Fig. 1A), in which one can see an almost complete degradation after 1 hour, the degradation of the Δ APNASC-FAT10 is strongly delayed.

In the text and figures the authors refer to the disappearance of ubiquitin conjugates as "ubiquitin degradation". Unlike FAT10, ubiquitin is typically not being degraded by the proteasome but is recycled. The authors may be referring here to degradation of proteins that are modified with ubiquitin instead of ubiquitin itself. However, the disappearance of ubiquitin conjugates cannot be used as a proxy for degradation of ubiquitylated proteins as their disappearance can also be due to

deubiquitylation of proteins. Instead, the authors can address this by analyzing the levels of a substrate for ubiquitin-dependent proteasomal degradation.

Reply: We agree with the reviewer that “ubiquitin degradation” is not the correct phrasing for the disappearance of poly-ubiquitinated proteins. Therefore, rephrased in the revised manuscript to “degradation of poly-ubiquitinated proteins”.

Fig 3. It is difficult to draw conclusions from this experiment. The effect on ubiquitin conjugates in the quantification is modest. Having this as a positive control, it is hard to say whether there is an effect on FAT10 turnover. If the authors on top of that argue in the discussion that the discrepancy between the data with siRNA transfection in Fig 3 and the inducible depletion in Fig 5 are due to the conditions with the siRNA being suboptimal I see little reason for including those experiments in the manuscript.

Reply: We agree with the reviewer that the information obtained from Figure 3 is rather limited. However, apart from showing that FAT10 20S proteasome degradation in vitro with purified components is different from degradation of FAT10 in cellulo, we also intend to illustrate in this manuscript, that the chosen method to analyze 20S proteasome degradation is crucial. Therefore, we decided to keep Figure 3 in the revised manuscript, also to illustrate the path how we reached the above conclusion.

Fig 4D. Why is the control AK cell line not included in this experiment?

Reply: Cell line AK is not a control cell line but a different clone that was generated by stable transfection of the miRNA encoding plasmid. Due to clonal variation in the AK cell line, Rpt2 knockdown was not that efficient (Figure 4C). Consistently, poly-ubiquitin accumulation was not as prominent as in the AU and AY clone. Therefore, clone AK was not used in further experiments. We mention this in the revised manuscript.

The authors write "... FAT10ylated substrates can be directly grabbed by 20S proteasomes in vitro...". This anthropomorphism may be misplaced as the 20S proteasomes lacks subunits that actively engage in protein recruitment.

Reply: We rephrased this sentence.

March 21, 2023

RE: Life Science Alliance Manuscript #LSA-2022-01760R

Dr. Michael Basler
University of Konstanz
Department of Biology, Division of Immunology
Universitätsstrasse 10
Konstanz 78457
Germany

Dear Dr. Basler,

Thank you for submitting your revised manuscript entitled "The ubiquitin-like modifier FAT10 is degraded by the 20S proteasome in vitro but not in cellulo". We would be happy to publish your paper in Life Science Alliance pending final revisions necessary to meet our formatting guidelines.

- please upload your supplementary figures as single files and add your supplementary figure legends to the main manuscript text
- please upload your table files as editable doc or excel files or make sure the tables are included in the doc file of your manuscript text
- please add a summary blurb/alternate abstract to our system
- please add the Twitter handle of your host institute/organization as well as your own or/and one of the authors in our system
- please use the [10 author names, et al.] format in your references (i.e. limit the author names to the first 10)

A. FINAL FILES:

B. MANUSCRIPT ORGANIZATION AND FORMATTING:

Sincerely,

Reviewer #1 (Comments to the Authors (Required)):

This is a revised manuscript in which the degradation of FAT10 in cellulo is shown to be dependent upon intact 26S proteasome whereas in vitro FAT10 is readily degraded by the 20S proteasome. It also presents a method of readily inactivating the 26S proteasome in cellulo by siRNA targeting of the PSMC1 gene. The comments of the reviewers have in my opinion been addressed in detail by the authors and the revised manuscript incorporates the suggested changes. Improvements in the revised manuscript compared to the original version render it acceptable for publication without further revision.

Reviewer #2 (Comments to the Authors (Required)):

The authors have addressed all the comments I raised in my earlier review, even though I still believe it might have been helpful to compare the efficiency of in vitro degradation of FAT10 by 20S and 26S proteasomes.

March 23, 2023

RE: Life Science Alliance Manuscript #LSA-2022-01760RR

Dr. Michael Basler
University of Konstanz
Department of Biology, Division of Immunology
Universitätsstrasse 10
Konstanz 78457
Germany

Dear Dr. Basler,

Thank you for submitting your Research Article entitled "The ubiquitin-like modifier FAT10 is degraded by the 20S proteasome in vitro but not in cellulo". It is a pleasure to let you know that your manuscript is now accepted for publication in Life Science Alliance. Congratulations on this interesting work.

DISTRIBUTION OF MATERIALS:

Again, congratulations on a very nice paper. I hope you found the review process to be constructive and are pleased with how the manuscript was handled editorially. We look forward to future exciting submissions from your lab.

Sincerely,
